# Comparing the reliability of relative bird abundance indices from standardized surveys and community science data at finer resolutions

**Mei-Ling Emily Feng***, Judy Che-Castaldo

Alexander Center for Applied Population Biology, Lincoln Park Zoo, Chicago, Illinois, United States of America

\* mefeng7@gmail.com

**Data Availability Statement:** All data and analysis code files are available from Figshare (https://doi.org/10.6084/m9.figshare.13477077).

## Abstract

Biodiversity loss is a global ecological crisis that is both a driver of and response to environmental change. Understanding the connections between species declines and other components of human-natural systems extends across the physical, life, and social sciences. From an analysis perspective, this requires integration of data from different scientific domains, which often have heterogeneous scales and resolutions. Community science projects such as eBird may help to fill spatiotemporal gaps and enhance the resolution of standardized biological surveys. Comparisons between eBird and the more comprehensive North American Breeding Bird Survey (BBS) have found these datasets can produce consistent multi-year abundance trends for bird populations at national and regional scales. Here we investigate the reliability of these datasets for estimating patterns at finer resolutions, inter-annual changes in abundance within town boundaries. Using a case study of 14 focal species within Massachusetts, we calculated four indices of annual relative abundance using eBird and BBS datasets, including two different modeling approaches within each dataset. We compared the correspondence between these indices in terms of multi-year trends, annual estimates, and inter-annual changes in estimates at the state and town-level. We found correspondence between eBird and BBS multi-year trends, but this was not consistent across all species and diminished at finer, inter-annual temporal resolutions. We further show that standardizing modeling approaches can increase index reliability even between datasets at coarser temporal resolutions. Our results indicate that multiple datasets and modeling methods should be considered when estimating species population dynamics at finer temporal resolutions, but standardizing modeling approaches may improve estimate correspondence between abundance datasets. In addition, reliability of these indices at finer spatial scales may depend on habitat composition, which can impact survey accuracy.

**Funding:** M.-L.E.F. and J.P.C. were supported by the National Science Foundation's Harnessing the Data Revolution program, award #1940276, https://nsf.gov/. The funder had no role in study design, data collection and analysis, decision to publish, or preparation of the manuscript.

**Competing interests:** The authors have declared that no competing interests exist.

## Introduction

Biodiversity loss is a global adversity in ecology [1], and can manifest as local extirpation, population declines and instability, species extinction, and changes in community composition [2–4]. Species abundance is a population measure that can be used to monitor many of these aspects of biodiversity loss and indicate risks to ecosystem health [3]. The relationship between biodiversity and the health of our environment makes biodiversity loss both a driver of and response to the risk of catastrophic events within coupled human-natural systems [2, 4–6]. With the increasing prevalence of natural disasters and human disturbance [4, 7], investigating the spillover of risks across human-natural systems is a growing field of research [8–11]. Amidst the "data revolution" with unprecedented data coverage across natural and anthropogenic domains [12], a major challenge is integrating heterogeneous spatiotemporal data to better understand the interactions between species populations and changes in other parts of human-natural systems [11].

Oftentimes, due to legislative and political boundaries and resource limitations, species monitoring efforts are limited to small scale studies in developed countries and regions, resulting in coarse resolution and discontinuous data [13]. Some of the most well-known broad scale population studies of birds in North America are limited to annual surveys, including the North American Breeding Bird Survey (BBS) [14], Audubon Christmas Bird Count [15], and Breeding Bird Atlas Surveys [16]. In particular, BBS is the largest and most comprehensive source of bird abundance data in North America, and is considered a reliable dataset for estimating bird population trends over time [2, 17–19].

Conversely, community science databases compile opportunistic observation data that can provide greater spatial coverage and finer spatiotemporal resolutions. eBird is a global community science network that has been collecting bird count and observation data since 2002 and has been growing in participation and data availability in recent years [20]. Due to its acceptance of historical survey data, a smaller subset of observations preceding the 2000s are also available. Data are collected through "checklists," which have fine spatial resolution, global coverage, and capture finer resolution temporal patterns [21]. Whereas BBS presents estimates of relative abundance (number of birds per route), eBird checklists are generally reduced to presence/absence, or "detection" data due to gaps in available count data. Making inferences from monitoring data requires standardized data collection methods and thorough sample sizes to account for variation in effort and detection probability. eBird uses semi-structured community science techniques with established methodologies for accounting for bias and effort within the data [22]. Its use of complete checklists allows for the assumption of true non-detections, and metadata on survey effort allow users to standardize the data and account for variation in detection probability.

eBird detection data have been previously used as proxies of relative abundance to estimate population trends across years, similar to those based on BBS data [18, 19, 21, 23]. These studies showed strong, positive correlations between the multi-year trends derived from eBird and BBS at the national level and within large provincial regions [18, 19]. However, a species' response to environmental change can occur at finer spatial and temporal resolutions, requiring data at finer scales to detect those critical changes. The goal of this study was to report the reliability of using these datasets to infer trends in bird abundances at finer spatial and temporal scales, using a case study of species within the state of Massachusetts. Specifically, we examined the consistency between relative abundance indices based on eBird and BBS data across statewide multi-year trends to verify the correspondence in trends found at coarser resolutions in previous studies [18, 19, 21, 23], and then at two finer temporal resolutions (annual estimates and changes in annual estimates between years) and two spatial resolutions (statewide

and within Massachusetts towns). For our index comparisons, we used the eBird and BBS datasets and two eBird modeling approaches to test whether using 1) different modeling methods and 2) standardized versus semi-structured surveys impacts the consistency of relative abundance indices at these different spatial and temporal resolutions. If these datasets provide consistent estimates that reliably reflect population changes at finer scales, it may allow for more flexibility when integrating with data from other domains to better understand changes and disturbances in human-natural systems.

## Materials and methods

### Study region and species

We focused on the state of Massachusetts from 2005–2018 as our study region and time period. eBird data are primarily available since 2005 [24]. Massachusetts has historically well-studied bird life and survey coverage across the state. The Massachusetts Division of Fisheries and Wildlife has been collecting records and monitoring populations of game, rare and endangered bird species since 1978 and Mass Audubon conducted the first North American Breeding Bird Atlas [25]. The engagement of community scientists in protecting native birds has only been growing in the last decade, with over 180,000 eBird checklists submitted for Massachusetts in 2018, giving the state the 8th highest checklist availability in the US states behind the leading states of California, Texas, and Florida, despite its smaller geographic area. The state is home to some of the largest designated Important Bird Areas (IBAs) in New England and encompasses grassland, freshwater wetland, saltwater marsh, forest, high intensity development, and coastline habitat. Massachusetts shoreline and off shore habitats are especially important stopover sites for pre- and post- bird migrations along the Atlantic coast and support nesting, feeding, and roosting for endangered coastal species [25, 26].

We chose 14 focal species known to be susceptible to causing electrical outages [27] for a related analysis. As such, we mean to present our findings as a case study applying eBird data rather than an assessment of the eBird dataset as a whole. Our species include cavity nesting species, species that travel and nest in large flocks, woodpeckers that damage the integrity of utility poles, and raptors which nest and perch near lines [28]. We did not consider species at the edge of their range or that did not have their year-round or breeding range within Massachusetts to ensure the data captured the state's regional breeding populations rather than transient migrants. Nocturnal species were not included because eBird surveys start after dawn while BBS surveys start just before dawn, making nocturnal species non-comparable between the two databases. All species occur on greater than 10% of eBird checklists in the study area to ensure a strong enough sample size for modeling. They include House Sparrow (*Passer domesticus*), European Starling (*Sturnus vulgaris*), Turkey Vulture (*Cathartes aura*), Mourning Dove (*Zenaida macroura*), Pileated Woodpecker (*Dryocopus pileatus*), Downy Woodpecker (*Picoides pubescens*), Hairy Woodpecker (*Leuconotopicus villosus*), Northern Flicker (*Colaptes auratus*), Red-bellied Woodpecker (*Melanerpes carolinus*), Red-tailed Hawk (*Buteo jamaicensis*), Osprey (*Pandion haliaetus*), Red-winged Blackbird (*Agelaius phoeniceus*), Brown-headed Cowbird (*Molothrus ater*), and Common Grackle (*Quiscalus quiscula*). Despite the limited sample size of our focal species, they cover a broad spectrum of detectability and prevalence within the state (Table 1), functional distinctness based on life history traits such as nesting habitat (cavities, trees, shrubs, cliffs), foraging strategy (ground, bark, soaring, diving), and diet (insectivores, carnivores, scavenger, and granivores), conservation status (invasives such as European Starling and conservation successes such as Osprey), and occupy a variety of breeding habitats (forest interior, coastal and open water, open habitat, and developed land).

**Table 1. Prevalence of 14 bird species on standardized Massachusetts eBird checklists and BBS transect stops during the 2005–2018 breeding seasons.**

| Family | Species | Number of Detections | | Detection Rate | |
|---|---|---|---|---|---|
| | | BBS | eBird | BBS | eBird |
| Icteridae | M. ater | 1,419 | 11,933 | 0.115 | 0.354 |
| | Q. quiscula | 2,024 | 21,698 | 0.164 | 0.568 |
| | A. phoeniceus | 2,062 | 22,725 | 0.167 | 0.609 |
| Picidae | D. villosus | 257 | 6,872 | 0.021 | 0.237 |
| | D. pubescens | 1,266 | 15,475 | 0.102 | 0.418 |
| | C. auratus | 476 | 12,898 | 0.038 | 0.363 |
| | D. pileatus | 188 | 2,860 | 0.020 | 0.173 |
| | M. carolinus | 897 | 10,401 | 0.073 | 0.340 |
| Accipitridae | B. jamaicensis | 92 | 8,493 | 0.008 | 0.264 |
| Pandionidae | P. haliaetus | 75 | 7,117 | 0.015 | 0.281 |
| Cathartidae | C. aura | 65 | 6,431 | 0.008 | 0.218 |
| Sturnidae | S. vulgaris | 1,344 | 12,686 | 0.109 | 0.380 |
| Passeridae | P. domesticus | 1,734 | 13,906 | 0.109 | 0.427 |
| Columbidae | Z. macroura | 4,009 | 22,057 | 0.325 | 0.557 |

The total number of surveys (eBird checklists and BBS stops) each species was detected on and the detection rate (proportion of surveys with detections out of total surveys ran) of the species, indicators of species prevalence, from 2005–2018.

## Data preparation

**eBird.** All data preparation and analysis were performed in R statistical language version 4.0.2 [29]. We first compiled and cleaned raw eBird checklist and sampling effort data from the eBird Basic Dataset (EBD; May 2019 Update) [30] to generate a set of detection data for each species. We filtered the eBird data with the R-package *auk* [24] to only include breeding season (May 1 through July 15) surveys, comparable with BBS protocol. We used additional auk filtering options to standardize survey effort to 10 surveyors or less, durations of 5 hours or less, travel distances of 5km or less, "stationary", "traveling", or "random" protocol types, and checklists marked as complete in order to generate detection data [24]. "Random" protocol types were reclassified as "traveling" if the corresponding survey distance traveled was greater than 0 km and "stationary" if distance traveled was listed as 0 km [19]. We removed checklists with less than five species reported as these often represent a targeted search and can affect model results [18, 19].

We filtered checklist locations that were within each species' observed range by only including surveys that fell within locations ("location_id") where the species had at least one observation within the study period [18]. We zero-filled checklists to create presence-absence data for each species at all checklists conducted within the range of its survey locations during the study period, reclassified sub-species and hybrids to the species level, and compiled duplicate observations made by group surveys into one unique checklist using the auk_zerofill function.

The eBird data Best Practices also recommend sub-sampling the data to reduce spatial and temporal bias, to balance the proportion of detections to non-detections, and balance sample sizes across years [22, 31]. More details on how we incorporated these data balancing methods in our data pre-processing can be found in S1 Appendix.

List length, or the number of species detected on a checklist has been used as an effective proxy for observer skill and effort when modeling populations with eBird data. It has been shown to capture variability caused by all of the effort covariates included in the eBird

metadata (e.g., number of observers, survey duration, time observations began, and length of travel) [18, 19]. Since each observation row in the data is a single species reported within a checklist, species observations from the complete EBD dataset were grouped by checklist_ids and counted to generate the number of species observed per checklist, or "list length."

**Breeding Bird Survey.** We compiled and filtered raw BBS data using methods similar to the eBird data in order to generate comparable detection data. BBS data were extracted from the North American Breeding Bird Survey Dataset 1996–2018, version 2018.0 [32]. We used raw 50-stop breeding bird data and non-breeding bird data filtered for Massachusetts. We grouped subspecies into species and filtered for high quality observations based on the "Run-Type" field indicating if the survey protocol followed USGS standards. The "routedataID" field was treated as an equivalent to eBird's "checklist_id", a unique run of a route, while the "Route" number was treated as eBird's "locality_id", a unique survey location. We filtered all records to include only routes for each species within its observed range.

Detection data for each species was calculated at each of the 50 stops along a run of a route to generate data comparable with eBird detection data [19]. Stops with species counts were given values of 1 to indicate species detections. If the species was not found at a stop along a route in a particular year, a record for that species was added with a 0 value to represent non-detections. The resulting BBS detection data for each species contained one survey observation per row, with each row being a stop within a survey route run in a particular year. These data were well-balanced across years, and we did not need to apply additional sampling to balance annual sample sizes.

Both cleaned eBird and BBS detection datasets were spatially joined to Massachusetts town names by their survey locations. Massachusetts towns were used as a consistent location identifier for both datasets. Table 1 shows each species' prevalence in each cleaned dataset.

## Modeling annual relative abundances

For our comparisons, we calculated four indices of annual relative abundance from our eBird and BBS datasets following existing protocols. Our two eBird indices were detection probability calculated using a generalized linear mixed model (GLMM) method [18, 19] and a random forest (RF) approach [21, 22]. Our two BBS indices were detection probability from the presence-absence data and relative abundance from the count data, both estimated using GLMMs [18, 19]. Comparing the two BBS indices examined whether detection probability was a reliable proxy for relative abundance (individual counts). Comparing the two eBird indices examined the reliability of two different modeling methods that estimate species detection probability. Finally, comparing detection probability between both eBird indices and BBS examined the reliability of estimates derived from different datasets as well as the influence of modeling methods on this reliability.

**eBird detection probability—GLMM method.** For our first eBird method, we calculated detection probability for each species using binomial GLMMs with the package *lme4* [19, 33]. We used a modified Laplace approximation method in which the random effects and fixed-effects coefficients are optimized in the penalized iteratively reweighted least squares step by setting the number of adaptive Gauss-Hermite quadrature points to zero (nAGQ = 0) [34, 35].

We compared model combinations that included day of year as a quadratic term, time of day surveys were started, list length, the natural log of list length, as well as additional effort variables (e.g., survey distance, duration, and number of observers). Models that ranked highest (based on lowest log likelihood) used quadratic effects of day of year and used only list length instead of multiple effort variables. Models using list length and the natural log of list length varied in rank between species and were selected on a species by species basis. We

ultimately fit models of detection probability based on the fixed effects of list length, protocol type, and the interaction of list length and protocol type to account for observation effort. The interaction between list length and protocol type follows the assumption that changes in sampling methods will change effort as well as species detection rates. We fit year as a factor to generate separate estimates for each year rather than an overall trend across years [14]. We used the day of year as a continuous, quadratic term to account for detection variability due to changes in species activity within the breeding season. Day of year and time of day were centered so both predictors had means of zero. We included checklist location (Massachusetts towns) and observer ID as random intercept terms.

We then used the resulting model to estimate Massachusetts town-level annual relative abundance for each species from 2005–2018. To generate predicted estimates, we used the time of day and day of year with the highest detection frequency, and the mean list length for each species, as well as traveling protocol type as input data to simulate a standardized survey conducted at the optimal time to detect the species. We set the random effect of observer ID to zero and predicted town-level annual estimates.

**eBird detection probability—RF method.** For this method we estimated detection probabilities defined by Johnston et al. (2020) and Strimas-Mackey et al. (2020) as the probability of encountering a species on a standardized eBird checklist [22, 31] (See S1 Appendix for detailed methodology). We first split the eBird data into testing (20%) and training data (80%), and used a balanced RF model (R-package *ranger* [36]) to model species detection as a function of temporal covariates (year, day of year, and starting time of observations) and list length to account for survey effort. Similar to previous studies [18, 19], we found list length was the most important effort predictor variable across all species in the RF (S1 Appendix). Towns were included to account for spatial variation in detection rates. We calibrated the estimated detection probabilities with observed detection rates using generalized additive models (GAM) fit with observed detection rates as the response to predicted detection probabilities. Using our reserved testing data, we found fair model performance across all species (averaged across all species models, AUC = 0.85, maximized Kappa = 0.51). Calibrating the models with the GAMs reduced the mean squared error from 0.15 to 0.13 (averaged across all species models). Town-level breeding season detection probabilities were predicted for each species by using an annual, town-level prediction dataset from 2005–2018. Prediction data variables were standardized by using the mean list length, traveling protocol, and the time of day and day of each month with maximum detection frequency for each species.

**BBS detection probability and relative abundance—GLMM method.** We built two more comparable GLMMs to the eBird model, but using BBS detection and count data to estimate state- and town-level, relative abundance indices for each species. We used a quasi-poisson distribution to fit the count data and a binomial distribution for the detection data. We did not include fixed effects for survey effort because BBS surveys use standardized methods that already account for survey effort. We used a fixed effect of year as a factor and random intercepts of town and observer. We account for any additional variation in observer skill by treating observer ID as a random effect. We predicted across the random effect of observer ID by setting its effect to zero to estimate town-level relative abundance. The resulting detection probability estimates were the probability of detecting a species at a random stop along a survey route in the state or in each town. The relative abundance estimates were the average number of individuals counted at a random stop along a survey route in the state or in each town.

We recognize these model-estimated relative abundance indices are associated with uncertainty, and that uncertainty likely differs across modeling approaches, species, and datasets. In order to account for these uncertainties in subsequent analyses, we calculated the standard errors (SE) for each estimate and used the inverse of these SEs as weights to place greater

weight on estimates with less uncertainty. We generated SEs for the GLMM model outputs (eBird and BBS detection probability and BBS relative abundance) via bootstrapping using the function *bootMer* and 200 simulations for each annual GLMM estimate. For RF model estimates we simply extracted the SEs from the model predictions. State-level annual estimates were calculated as the weighted average of town-level estimates in each year, with weights being the inverse SE of each town-level estimate.

## Comparing modeling methods and datasets at finer resolutions

To quantify the reliability between relative abundance indices, we calculated correlation coefficients between each within- and between-dataset pair of indices and assessed their significance using the *p*-values extracted from the *corr.test* function. For each state-level pairwise comparison, we calculated Pearson correlation, or Spearman-Rank correlation when estimates were not normally distributed (Shapiro-Wilk $p < 0.05$). For town-level index comparisons, we used a weighted Pearson correlation using the *cor.wt* function from R-package *psych*. We weighted the matrix of index annual estimates by the inverse SEs for each annual estimate. We first compared relative abundance indices within datasets that used different methodologies. We assessed whether transforming BBS count data into detection data retained similar estimates of relative abundance. This was predominantly to ensure that detection data was a proxy of abundance, allowing for meaningful comparisons across all of our indices. We then compared the consistency of our two eBird modeling methods (GLMM- and RF-estimated detection probabilities) to see if these two approaches yielded similar results. Our second set of comparisons were between datasets to assess whether differences between eBird and BBS survey structure affected the precision of their abundance estimates. We tested the correlation of each BBS index with each eBird index to see whether modeling methods affected this between-dataset precision.

We assessed how these correlations between our four relative abundance indices changed across spatiotemporal resolutions. We first confirmed previous findings by examining the consistency in multi-year, statewide trends between indices to compare how the consistency of indices at this coarser resolution compared to finer spatiotemporal resolutions. We visualized multi-year trends by plotting the time series of annual estimates for each index with two trend lines, similar to [19]: a LOESS smooth with a span of two and the slope from binomial generalized linear models (GLMs) fit to the annual estimates across years. To quantitatively test correspondence of these multi-year trends, we evaluated whether the slope and 95th percent confidence intervals of each index's GLM coefficients overlapped. For these GLMs, we standardized the four relative abundance indices, centered the year variable, and included a quadratic term for year to account for non-linear patterns observed.

To test how consistent these indices were at finer temporal resolutions, we calculated the above mentioned correlations between each index's statewide annual relative abundance estimates and the inter-annual changes in these estimates. We calculated inter-annual changes as the difference between the annual estimate in the previous year and the present year as a proportion of the estimate in the previous year. We visualized correspondence between indices by plotting their annual estimates and inter-annual changes in side-by-side time series. Additionally, we used box plots to compare the distributions of correlation coefficients across species for each pairwise index comparison of annual estimates and inter-annual changes. This compared the overall correlation between each index at these two finer temporal resolutions across our species samples.

We next investigated the consistency of index annual estimates and inter-annual changes at finer spatial scales (by town) and whether between dataset index reliability was influenced by

local spatial attributes. Since BBS count and detection data yielded similar estimates, we only compared BBS detection probability against both eBird indices. We visualized how correlation varied across towns by mapping towns shaded by their correlation coefficients. We again used box plots to compare the distributions (using the complete set of observations across species and towns) of correlation coefficients between each comparison of eBird and BBS indices. To test whether any spatial attributes such as birding hot spots or habitat type influenced survey accuracy, and indirectly the precision of indices between eBird and BBS, we used a linear mixed model to predict correlation coefficients between indices based on town specific attributes, including the fixed effects of eBird checklist location density (a proxy for birding hot spots) and the area proportion of habitat types classified as aggregated land cover types (developed, wetland, herbaceous, deciduous forest, and shrub) from the National Land Cover Database 2016 release [37]. We scaled and centered all predictor variables. Since only the starting points of BBS routes had available spatial locations (n = 25), we did not have a complete sample of all towns within Massachusetts. To increase the sample size for the regression analysis, we combined the town-level correlation coefficients across species (319 observations) and included species as a random effect.

## Results

### Comparing indices within datasets at different temporal resolutions

For our within dataset index comparisons, GLMMs using BBS count data and detection data predicted statewide relative abundance estimates that were highly correlated for all 14 species (S2 Appendix) and this correlation was consistent at different temporal resolutions (Fig 1). At a coarser temporal resolution, multi-year trends remained consistent between the two BBS indices for all species except *Q. quiscula*, *D. villosus*, and *P. domesticus* (Table 2). For finer resolution of annual estimates, correlation (*r*) between each species' estimates ranged between 0.61–1.00, $p < 0.05$. Species also had strong positive correlations between the inter-annual changes of these two BBS indices (*r* ranged between 0.6–1.00, $p < 0.05$) with the exception of *S. vulgaris* which had the weakest correspondence between the inter-annual changes in its BBS indices (*r* = 0.41, $p > 0.05$).

In comparison to BBS, within dataset consistency between the two statewide eBird indices was weaker and correlation further decreased at finer resolutions (Fig 1). Nine of the 14 species had consistent multi-year trends between eBird indices (Table 2), but at the finest temporal resolution, the eBird indices showed no significant, positive correlations between inter-annual changes with the exception of *D. pubescens*. See S2 Appendix for correlation matrices of the within dataset indices at annual and inter-annual resolutions for each species.

### Comparing indices between datasets at different temporal resolutions

At the coarsest temporal resolution of multi-year trends, relative abundance indices were relatively consistent between datasets, but this consistency was not found across all of our study species. 12 species (85%) had consistent multi-year trends between at least one index derived from each dataset (Table 2) (S3 Appendix). Two of these species, *D. pileatus* and *M. carolinus*, had consistent and significant trends across all four indices (Fig 2). At the annual resolution, these two species in addition to *D. villosus*, *B. jamaicensis*, *P. haliaetus* and *P. domesticus* (six total species) had strong positive correlations (*r* > 0.4) for all between-dataset index comparisons (Fig 3). Five more species (a total of 11 species [79%]) showed strong positive correlations between annual estimates of at least one BBS and eBird index (Table 3). Over all of our study species, the ebird GLMM-DP method had greater consistency with BBS DP than the RF method at the annual resolution (Fig 1). However, this correspondence diminished when

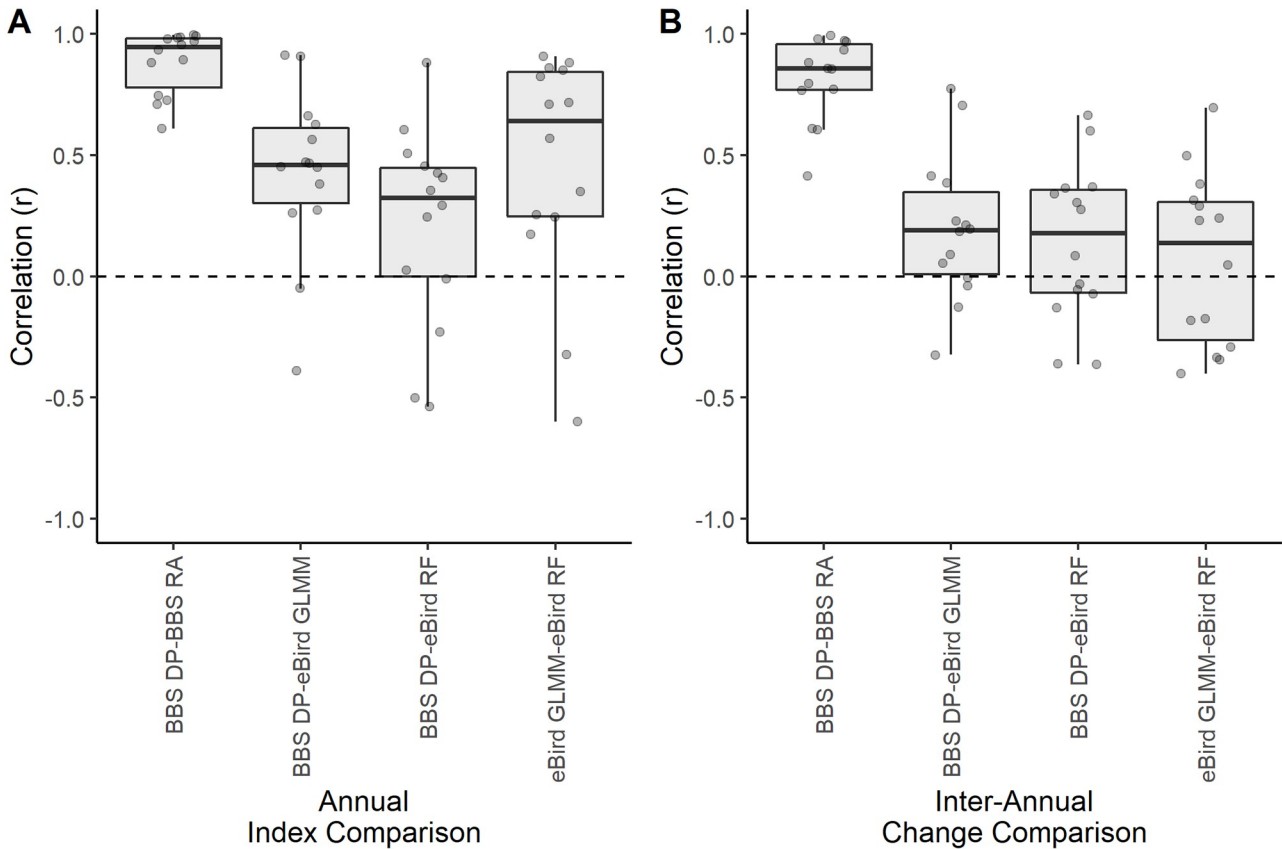

**Fig 1. Distribution of correlation coefficients comparing four statewide relative abundance indices for 14 bird species.** The box plots show the distribution of 14 bird species' correlation coefficients comparing A) the annual estimates and B) inter-annual changes between pairs of four statewide relative abundance indices: BBS relative abundance from detection and count data (BBS DP-BBS RA), BBS detection probability and eBird GLMM detection probability (BBS DP-eBird GLMM), BBS detection probability and eBird RF detection probability (BBS DP-eBird RF), and eBird detection probability from GLMMs and RF models (eBird GLMM-eBird RF).

looking at inter-annual changes of these estimates (Table 3) (S2 Appendix). We did not find strong between-dataset correspondence at inter-annual temporal scales, regardless of the eBird modeling approach used (Fig 1).

## Comparing fine resolution indices between datasets at a smaller spatial scale

Mapping the correlation between BBS and eBird annual estimates and inter-annual changes across Massachusetts towns showed that correlation between indices was mostly consistent across towns (S4 Appendix). Variation in town-level correlation was only found in the eBird RF and BBS detection probability comparison because we modeled towns as random inter-cepts for both the eBird and BBS GLMM estimates, which resulted in the same year to year population changes for all towns. The spatial patterns and magnitude of index correlations changed across temporal resolutions and species. However, even at this finer spatial scale, annual estimates of eBird and BBS indices for *D. pileatus* and *M. carolinus* (S4 Figs 10, 11 in S4 Appendix, respectively) have high correspondence that is consistent across towns.

We used a linear mixed model to examine whether correspondence between eBird and BBS indices depended on habitat type and eBird survey density. While accounting for variation

**Table 2. Multi-year trends (2005–2018) and their 95th percent confidence intervals for Massachusetts populations of 14 study species.**

| Family | Species | BBS DP | BBS RA | eBird DP-GLMM | eBird DP-RF |
|---|---|---|---|---|---|
| *Icteridae* | *M. ater* | -0.09 (-0.21, 0.02) | -0.1 (-0.22, 0.02) | **0.13 (0.00, 0.26)** | **0.20 (0.11, 0.29)** |
| | *Q. quiscula* | -0.12 (-0.26, 0.02) | **-0.13 (-0.26, -0.01)** | **-0.15(-0.27, -0.02)** | 0.06 (-0.08, 0.21) |
| | *A. phoeniceus* | -0.02 (-0.18, 0.14) | -0.01 (-0.16, 0.15) | -0.04 (-0.18, 0.11) | **-0.15 (-0.22, -0.09)** |
| *Picidae* | *D. villosus* | 0.12 (-0.01, 0.25) | **0.13 (0.01, 0.26)** | **0.19 (0.10, 0.28)** | **0.20 (0.14, 0.27)** |
| | *D. pubescens* | 0.07 (-0.08, 0.21) | 0.06 (-0.09, 0.21) | **0.09 (-0.05, 0.23)** | **0.14 (0.03, 0.25)** |
| | *C. auratus* | 0.10 (-0.04, 0.25) | 0.11 (-0.03, 0.25) | 0.11 (-0.03, 0.25) | **0.15 (0.10, 0.21)** |
| | ***D. pileatus*** (P) | **0.14 (0.02, 0.27)** | **0.14 (0.02, 0.27)** | **0.22 (0.17, 0.27)** | **0.21 (0.17, 0.26)** |
| | ***M. carolinus*** (P) | **0.22 (0.16, 0.28)** | **0.22 (0.17, 0.28)** | **0.23 (0.18, 0.27)** | **0.17 (0.09, 0.25)** |
| *Accipitridae* | *B. jamaicensis* | **0.14 (0.03, 0.25)** | **0.14 (0.03, 0.25)** | 0.10 (-0.02, 0.23) | **0.16 (0.05, 0.28)** |
| *Pandionidae* | *P. haliaetus* | **0.13 (0.00, 0.28)** | **0.13 (0.00, 0.28)** | **0.18 (0.08, 0.28)** | 0.01 (-0.15, 0.17) |
| *Cathartidae* | *C. aura* | 0.12 (-0.01, 0.26) | 0.09 (-0.05, 0.24) | **0.20 (0.11, 0.28)** | **0.10 (0.02, 0.18)** |
| *Sturnidae* | *S. vulgaris* | -0.08 (-0.18, 0.02) | 0.05 (-0.04, 0.15) | 0.11 (-0.03, 0.24) | **0.19 (0.09, 0.29)** |
| *Passeridae* | *P. domesticus* | 0.03 (-0.12, 0.19) | **0.17 (0.05, 0.28)** | **0.19 (0.10, 0.29)** | 0.10 (-0.03, 0.24) |
| *Columbidae* | ***Z. macroura*** (NS) | 0.03 (-0.12, 0.17) | -0.03 (-0.17, 0.11) | 0.03 (-0.13, 0.19) | -0.04 (-0.19, 0.10) |

Multi-year trends are derived from the coefficients of linear models fit to standardized annual estimates of four relative abundance indices across years as a centered, quadratic polynomial. Species with names in bold showed consistent, positive (P) or non-significant (NS) trends including 95th percent confidence intervals across all indices from both datasets (Breeding Bird Survey [BBS] and eBird) and modeling methods (GLMM and RF). Trend values in bold indicate a directional, non-zero trend (positive or negative). Species are grouped into their respective taxonomic families.

DP = Detection Probability

RA = Relative Abundance (from count data)

across species, we found a significant relationship between index correspondence and habitat type when comparing annual BBS GLMM and eBird RF detection probability indices. Towns with higher proportions of herbaceous habitat (e.g., grasslands) showed greater correspondence between those indices (Table 4). Correspondence between inter-annual changes in BBS and eBird detection probability did not show significant relationships with our spatial predictors.

The distribution across species of correlation coefficients between town-level eBird indices and BBS detection probability were similar to the state level results. The eBird GLMM method was more strongly correlated with BBS detection probability across species at the annual resolution, and both eBird modeling methods showed little correspondence with BBS at the inter-annual resolution (Fig 4).

## Discussion

Reliable estimates of species abundance are needed to monitor biodiversity loss and changes in ecosystem health over time. Standardized abundance trends across multiple species are currently assessed at multi-year time scales, but finer resolutions are needed to better understand the impacts of disturbances in human-natural systems. We examined whether existing broad scale species data can be used to delineate finer resolution patterns by comparing multiple indices of relative abundances based on two bird occurrence datasets. Despite our small sample of focal species, they represent a wide range of species-specific factors known to influence detectability and data reliability including forage and nesting habitats, flocking behavior, and body size [38–42] and we find consistency between indices is variable across these species (Figs 1 and 4). It is possible that expanding this analysis to more species could help delineate species patterns. However, our results show diverse patterns even among species within a

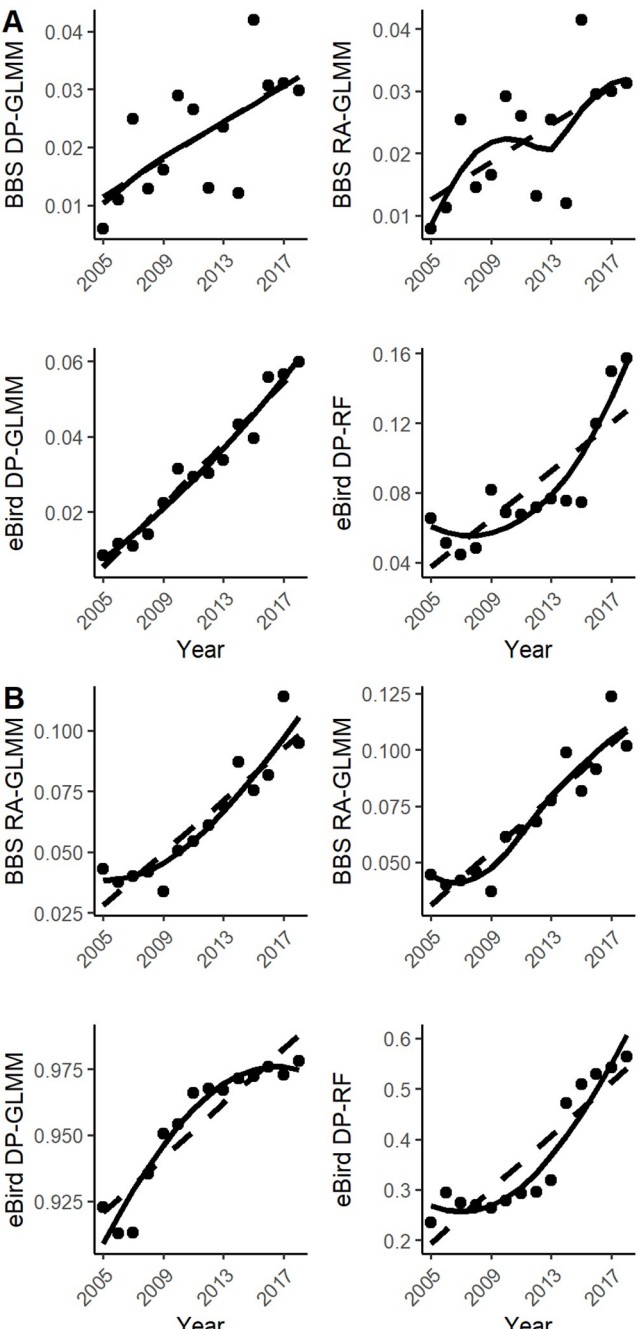

**Fig 2. Multi-year trends across four relative abundance indices.** Four indices of relative abundance within and between eBird and BBS datasets capture similar multi-year trends across A) Pileated Woodpecker (*Dryocopus pileatus*) and B) Red-bellied Woodpecker (*Melanerpes carolinus*). Trends are plotted as generalized linear regressions and LOESS smoothed lines fit through annual estimates over time.

taxonomic group (e.g., *Icteridae* and *Picidae*). It is likely that broadening the array of study species would only add to the diversity of results in index consistencies and overall conclusions would be similar.

BBS detection probability was a reliable proxy for relative abundance across temporal resolutions for our study species, with all correlations between BBS indices significant and

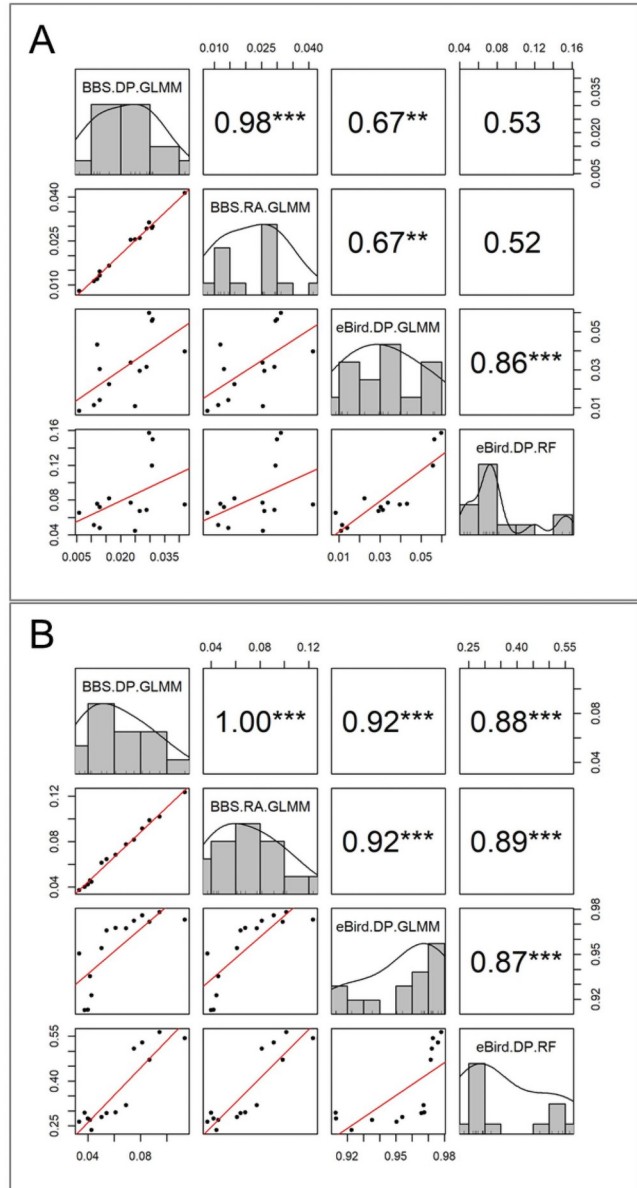

**Fig 3. Correlation matrices of annual relative abundance estimates based on four indices show correspondence within and between datasets.** Data distributions, as well as scatter plots and correlation coefficients are provided for each index of A) Pileated Woodpecker (*Dryocopus pileatus*) and B) Red-bellied Woodpecker (*Melanerpes carolinus*) annual relative abundance. There was strong correspondence across different indices within and between datasets (eBird and BBS), suggesting reliable estimates of relative abundance for these species. * indicates the strength of a significant correlation coefficient (Pearson's and Spearman's when data were non-normal).

coefficients ≥0.60, except for *S. vulgaris*. A possible explanation is that it is a flocking species and surveys for such species tend to have higher errors, lower accuracy, and underestimate true abundance because accurately counting several birds in a flock requires greater observational skill [42]. Similar flocking species including *A. phoeniceus* and *Q. quiscula* also had the next lowest correlations between annual estimates (*r* = 0.6). We also compared two well-cited methods for modeling eBird data [19, 31] that estimate the probability of detecting a species

**Table 3. Consistency of annual estimates and inter-annual changes between relative abundance indices derived from eBird and BBS datasets.**

| | | Annual Estimates | | | | Inter-Annual Changes | | | |
| --- | --- | --- | --- | --- | --- | --- | --- | --- | --- |
| | | eBird GLMM | | eBird RF | | eBird GLMM | | eBird RF | |
| Family | Species | BBS DP | BBS RA | BBS DP | BBS RA | BBS DP | BBS RA | BBS DP | BBS RA |
| *Icteridae* | *M. ater* | -0.05 | -0.09 | **-0.54** | **-0.54** | 0.09 | 0.05 | -0.03 | -0.01 |
| | *Q. quiscula* | 0.26 | **0.62** | 0.35 | 0.49 | 0.05 | **0.58** | 0.27 | 0.25 |
| | *A. phoeniceus* | 0.38 | 0.41 | 0.29 | **0.62** | 0.19 | 0.26 | 0.37 | **0.87** |
| *Picidae* | *D. villosus* | 0.47 | **0.53** | 0.43 | **0.53** | 0.19 | 0.10 | **0.66** | **0.79** |
| | *D. pubescens* | **0.63** | **0.55** | 0.24 | 0.17 | 0.21 | 0.15 | -0.05 | -0.10 |
| | *C. auratus* | 0.27 | 0.30 | -0.23 | -0.30 | -0.01 | -0.09 | **0.60** | **0.60** |
| | *D. pileatus* | **0.66** | **0.67** | 0.51 | **0.53** | -0.13 | -0.14 | -0.36 | -0.29 |
| | *M. carolinus* | **0.91** | **0.92** | **0.88** | **0.89** | -0.32 | -0.19 | -0.30 | -0.31 |
| *Accipitridae* | *B. jamaicensis* | 0.47 | 0.52 | 0.41 | 0.41 | 0.23 | 0.27 | -0.07 | -0.03 |
| *Pandionidae* | *P. haliaetus* | **0.91** | **0.85** | **0.60** | **0.70** | **0.77** | **0.76** | 0.34 | 0.45 |
| *Cathartidae* | *C. aura* | 0.45 | 0.35 | 0.02 | 0.24 | -0.04 | -0.24 | -0.13 | 0.15 |
| *Sturnidae* | *S. vulgaris* | -0.39 | -0.10 | -0.50 | -0.23 | 0.38 | 0.19 | 0.36 | -0.20 |
| *Passeridae* | *P. domesticus* | **0.56** | **0.79** | 0.45 | **0.66** | **0.70** | 0.50 | 0.08 | 0.03 |
| *Columbidae* | *Z. macroura* | 0.45 | 0.45 | -0.01 | -0.06 | 0.41 | 0.31 | -0.36 | -0.36 |

Correlation coefficients (Pearson's *r*, and Spearman's *r* when data was non-normal) show the consistency of annual estimates and the inter-annual changes between detection probability from two eBird methods (a GLMM and a RF model) and two BBS relative abundance indices. Species are grouped into their respective taxonomic families. Bold values indicate significant correlations (*p*<0.05).

DP = Detection Probability

RA = Relative Abundance (from count data)

on a survey (a proxy of relative abundance). Despite eight of our study species showing consistent annual estimates, only *D. pubescens* maintained consistent inter-annual changes across these two eBird indices (S2 Appendix). This lack of correspondence between different eBird modeling methodologies was unexpected, and stresses the importance of selecting the appropriate modeling method for the questions at hand and understanding their assumptions. Our results show that using consistent modeling methods (in our case, GLMMs) can increase the

**Table 4. Summary of the linear mixed model using weighted Pearson's correlation coefficients between annual BBS and eBird RF detection probability indices as a response to spatial attributes summarized within Massachusetts towns.** Reported are the model estimates and 95th percent confidence intervals. The spatial attribute predictors are eBird survey density, the number of survey locations per $km^2$ area of each town, and habitat types as the proportion of land cover area within each town. Town-level index comparisons were combined across 14 study species and species IDs were included as a random effect to account for species specific effects.

**Correlation Coefficient of eBird-BBS Indices**

| *Predictors* | *Estimates* | *95% CI* |
| --- | --- | --- |
| (Intercept) | 0.15 | (0.02, 0.28) |
| eBird Survey Density | -0.01 | (-0.04, 0.02) |
| Deciduous Forest | 0.01 | (-0.01, 0.02) |
| Shrub | 0.001 | (-0.03, 0.03) |
| Herbaceous | 0.04 | (0.01, 0.08) |
| Developed | 0.01 | (-0.03, 0.05) |
| Wetland | 0.002 | (-0.02, 0.02) |
| Observations | 319 | |
| Groups | 14 | |

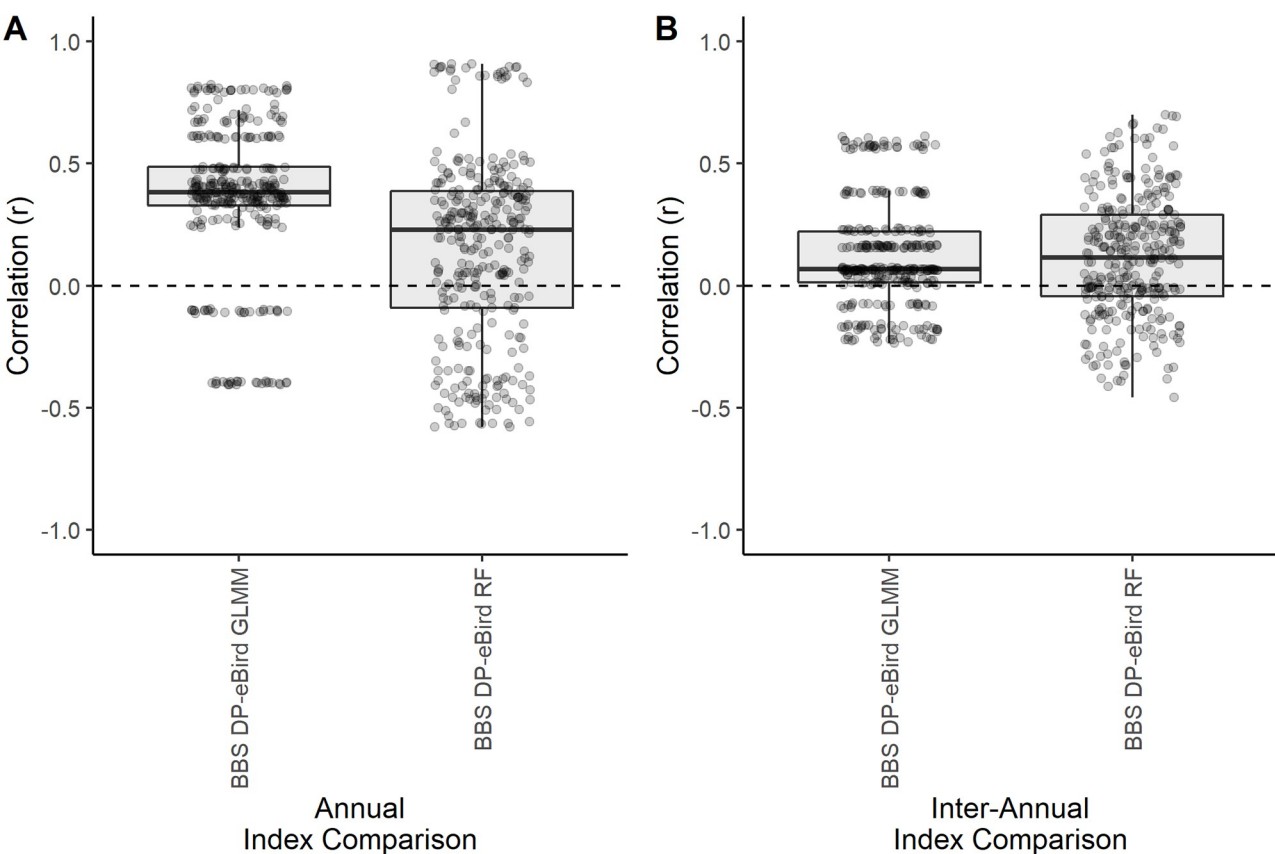

**Fig 4. Distribution of weighted Pearson's correlation coefficients for 14 bird species comparing town-level relative abundance indices between eBird and BBS.** Box plots show the distribution of 14 bird species' correlation coefficients comparing A) the annual estimates and B) inter-annual changes of town-level BBS detection probability (DP) with eBird GLMM-DP and RF-DP.

consistency between relative abundance estimates derived from different datasets. Different modeling methods (GLMMs and RFs) can alter abundance estimates even when using the same dataset, however each modeling method may offer advantages at different resolutions. GLMMs may result in more interpretable estimates at coarser resolutions [18, 19], while the random forest approach could be seen as more flexible [43] for predicting estimates at finer resolutions, allowing inclusion of additional predictors of species occurrence [35, 44].

In addition to inconsistent modeling approaches, we found that finer temporal resolutions also decreased the overall consistency between relative abundance estimates derived from eBird and BBS observation data. eBird data, once adjusted for effort and bias, have been shown to reliably capture multi-year population trends similar to structured surveys from the BBS at regional and national scales [18, 19, 21], although this reliability seems to decline at larger, global scales [45]. Expanding on these scale dependent findings, we show that multi-year trends are more consistent across different indices and databases than the finer temporal resolution annual estimates and inter-annual changes. Four of our 14 study species had multi-year trends that were consistent between eBird and BBS (Table 2), while eight additional species had consistent multi-year trends between at least one index derived from each dataset. This index agreement was maintained in the annual estimates, with 11 of 14 species having at least one significant correlation between indices across databases (Table 3). In particular, two species (*M. carolinus* and *D. pileatus*) had especially high correspondence across their eBird and

BBS annual relative abundance indices, which could be attributed to the strength of their population trends.

In the second Massachusetts Breeding Bird Atlas (BBA), completed in 2011, Red-bellied Woodpecker (*M. carolinus*) was the species with the second greatest increase in occupied breeding area in the state since the previous BBA in the 1970s. Pileated woodpecker (*D. pileatus*) also had significant increasing BBA trends in the state, consistent with long-term BBS trends [25, 46, 47]. This finding suggests there may be a minimum effect size in terms of change in abundance in order for the change to be consistently captured across datasets. Previous studies have shown relative abundance measures for highly prevalent and rare species are more difficult to predict because they over- and under-saturate the model with occurrences, respectively [19]. Other similar studies have shown niche specialization and species distribution also play roles in the overall detectability of a species and thus the accuracy of its relative abundance indices [18]. We suggest future studies examining a larger sample of species should consider how the long-term population trend and the strength of this trend affect the accuracy of eBird estimates.

Despite some correspondence between annual estimates, the inter-annual changes show less correspondence between eBird and BBS (Figs 1 and 4). Although the annual estimates for each year may be similar relative to the range of abundance estimates within the study period, the differences from one year to the next may be different in magnitude or direction depending on the dataset and index being used (see S3 Appendix for multi-year trend plots and time series for each index). An earlier study on the Wood Thrush (*Hylocichla mustelina*) [21] found positive correspondence between eBird and BBS inter-annual change when using a standardized start date and range-wide annual relative abundance indices. It is possible that our focus on a subset of the species range as delineated by state boundaries is not ecologically important, and correspondence between datasets may improve when the entire range of each species is considered. It is also possible that dynamics simply differ between species, as patterns differed among the species in our study as well.

Similarly, we found that correspondence between relative abundance indices at the finer, town-level spatial scale showed some variation between towns. However, the degree of this variation was inconsistent between species and temporal resolutions (S4 Appendix). We also found local spatial attributes such as habitat cover may affect the degree of correspondence between eBird and BBS abundance indices. This result is consistent with previous findings that habitat structure can affect observer capabilities by impacting the travel of wind and noise, and impairing visual observation [38, 40, 48, 49]. We note that our spatial analysis was limited due to the number of MA towns with BBS routes, and further study is needed to better understand the impact of habitat types and other spatial attributes on the reliability of these abundance estimates.

In absence of agreement among the indices, it is unclear which index or dataset is most accurate in representing species abundances at these finer temporal and spatial scales. This finding underlines the importance of validating eBird against structured surveys at finer temporal resolutions. However, the narrow temporal windows of broad scale, structured surveys [14–16] limits their ability to validate eBird data at intra-annual resolutions. In most cases, localized monitoring efforts are the best validation data available at these resolutions [50, 51]. Nevertheless, community science projects such as eBird compile finer resolution data at broad extents, opening opportunities for a more complete understanding of ecosystem health and diversity in response to finer scale drivers of change such as land use and infrastructure [52, 53]. Integrating species abundance with drivers of ecological change remains a complicated task due to heterogeneous scales and resolutions across data sources. Community science data

provides flexibility to consider ecological processes at organism-relevant scales that vary across species and environmental variables [53].

## Conclusion

The results of this case study highlight the need for increased vigilance as future bird population studies consider various modeling applications of community science data. By comparing relative abundance indices across structured and community science databases, we show that modeling methods, survey structure, and the spatiotemporal resolutions at which they are applied can all impact relative estimates of species abundance. All of our tested indices proved to be relatively reliable as proxies of abundance at the coarser scale of multi-year trends. At finer spatial and temporal scales however, relative abundance estimates derived from different datasets and modeling methods point to inconsistent pictures of abundance depending on the species. Therefore the reliability and appropriateness of these various indices depends on the scale and resolution of the analysis. Our findings indicate that multiple modeling methods and data sources should be tested and carefully considered when addressing questions that require species abundance trends at finer spatial and temporal resolutions.

## Supporting information

**S1 Appendix. eBird random forest detection probability modeling methods.** Details on the random forest method used to estimate the breeding season detection probability of 14 bird species in Massachusetts.
(ZIP)

**S2 Appendix. Correlation matrices of four relative abundance indices within and between datasets.** Matrices for 14 Massachusetts bird species comparing the correspondence of annual estimates and their inter-annual changes between relative abundance indices.
(PDF)

**S3 Appendix. Multi-year trends and annual time series plots for four relative abundance indices.** Multi-year trends (2005–2018) for 14 Massachusetts bird species. Trends are plotted from the annual estimates of each index smoothed by a LOESS curve and fit to generalized linear models. Time Series plots and linear model coefficients of the quadratic term of year for the annual estimates show the magnitude and directions of inter-annual changes for each index.
(PDF)

**S4 Appendix. Correlation between annual relative abundance indices and their interannual changes derived from two datasets within Massachusetts towns for 14 bird species.** Massachusetts town maps colored by their Spearman Rank Correlation coefficients of relative abundance estimates between eBird and BBS datasets.
(PDF)

**S1 File. eBird and BBS data and R-script for reproduction of all figures and tables.** The processed data from eBird and the BBS used in our analyses and the analysis code are available through figshare (https://doi.org/10.6084/m9.figshare.13477077). The R-script covers the data preparation and modeling for BBS and eBird data, as well as the analyses and plots for comparing the four relative abundance indices.
(CLS)

## Acknowledgments

We thank the volunteers and community scientists who conducted Breeding Bird Survey routes and contributed to the eBird database as well as the eBird project team. This work was made possible by the United States Geological Survey, the Canadian Wildlife Service of Environment Canada, and Cornell Lab of Ornithology who have provided open access to their datasets.

## Author Contributions

**Conceptualization:** Judy Che-Castaldo.

**Data curation:** Mei-Ling Emily Feng.

**Formal analysis:** Mei-Ling Emily Feng.

**Funding acquisition:** Judy Che-Castaldo.

**Investigation:** Mei-Ling Emily Feng, Judy Che-Castaldo.

**Methodology:** Mei-Ling Emily Feng, Judy Che-Castaldo.

**Project administration:** Mei-Ling Emily Feng, Judy Che-Castaldo.

**Resources:** Judy Che-Castaldo.

**Software:** Mei-Ling Emily Feng, Judy Che-Castaldo.

**Supervision:** Judy Che-Castaldo.

**Validation:** Mei-Ling Emily Feng, Judy Che-Castaldo.

**Visualization:** Mei-Ling Emily Feng, Judy Che-Castaldo.

**Writing – original draft:** Mei-Ling Emily Feng, Judy Che-Castaldo.

**Writing – review & editing:** Mei-Ling Emily Feng, Judy Che-Castaldo.

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
