## [Decision Letter · Decision Letter 0]

2 Feb 2021

PONE-D-20-40357

Reliability of relative bird abundance indices at fine-scale temporal resolutions

PLOS ONE

Dear Dr. Feng,

Thank you for submitting your manuscript to PLOS ONE. After careful consideration, we feel that it has merit but does not fully meet PLOS ONE’s publication criteria as it currently stands. Therefore, we invite you to submit a revised version of the manuscript that addresses the points raised during the review process.

We look forward to receiving your revised manuscript.

Kind regards,

Daniel de Paiva Silva, Ph.D.

Academic Editor

PLOS ONE

Additional Editor Comments:

Dear Feng et al.,

After independent reviews from two reviewers, I believe you manuscript will be suitable suitable for publication after you take care of the important issues raised by reviewer #1. If after your review, the reviewers agree that your text improved, I will be glad to accept it for publication. Considering the pandemic scenario, you will have 3 months to deliver your improved manuscript, by May 1st 2021. In case anything else requires further explanation, please let me know.

Sincerely,

Daniel Silva

Reviewers' comments:

Reviewer's Responses to Questions

**Comments to the Author**

1. Is the manuscript technically sound, and do the data support the conclusions?

Reviewer #1: Partly

Reviewer #2: Yes

2. Has the statistical analysis been performed appropriately and rigorously? 

Reviewer #1: Yes

Reviewer #2: Yes

3. Have the authors made all data underlying the findings in their manuscript fully available?

Reviewer #1: Yes

Reviewer #2: Yes

4. Is the manuscript presented in an intelligible fashion and written in standard English?

Reviewer #1: Yes

Reviewer #2: Yes

5. Review Comments to the Author

Reviewer #1: Multiple studies have compared standardized bird survey data with community science (eBird) data at coarse spatial scales. This study aims to compare these data sources at a finer scale. The study compares two Breeding Bird Survey indices with two eBird indices during the breeding season in Massachusetts. Correlation was highest between the two BBS indices, moderate between the two eBird indices, and relatively low among data sources, particularly for inter-annual changes in relative abundance.

I appreciate the ideas behind this study. In an era when vast amounts of community science data are becoming available it is critical to make comparisons with standardized data sources. For the most part I think the methods are suitable, although important clarifications are needed in several places. My main concerns are ones of scope, and the generality of the findings.

First, the focal species give cause for concern. They appear to have been selected for another, very different, study investigating birds that can cause electrical outages! This does not seem to be the best criterion for a study testing data reliability. The result is just 14 species from eight families. Five species are woodpeckers, three species are icterids, and three are raptors, showing high taxonomic bias. Two species are also invasive, which is interesting but never mentioned. To fully realize the aims of the study, this limited species list does not seem appropriate. To make reasonable inferences I would want to see many more species of higher taxonomic and ecological diversity. There are no study limitations that preclude the analysis of more species; the eBird data are available on all.

Second, the aims were to investigate reliability at finer spatio-temporal resolutions. I think the study does a good job at the temporal aspect. However, simply restricting the study to Mass in my opinion falls short of investigating spatial variation in reliability. I think it would be awesome to test reliability state-by-state, but I understand that might be too large in scope. But two ideas the authors could consider are assessing reliability by habitat, and assessing differences in reliability between towns. I can imagine a map of towns across Mass with each colored by the degree of correlation between indices. It brings to mind the paper by Jarzyna et al (2015, Global Ecol. Biogeogr.) looking at the spatial scaling of temporal changes in New York State.

Minor Comments:

The title seems a little vague; it doesn’t really give a sense of the paper. Perhaps something like: “Comparing the reliability of bird abundance indices between standardized surveys and community science data at fine-scale temporal resolutions”.

L31: The fact that eBird data can capture intra-annual changes in abundance is mentioned a couple of times, but how can one compare intra-annual abundance changes between community science data and standardized survey data? The latter have narrow temporal windows (the breeding season, or Christmas). This is not really an issue the paper addresses.

L58: Why not 2005-2020? eBird data have increased exponentially, so two years could make a big difference. Moreover, when estimating temporal trends over such short timespans, every year counts!

L64: Again, this number was surely much larger in 2020.

L68-69: How does the reliability of abundance indices vary by habitat?

L73: 14 is not many species and, given the aims of this paper, the subset of species chosen (those that can cause electrical outages) does not seem relevant or representative.

L101: This criterion might not be stringent enough. Even if I’m looking at a target species, I might add a house finch or starling as it flies over. Perhaps ≥5 would be a better cut-off. Horns et al. (reference #18) use ≥4.

L135-136: Geographically, did BBS survey locations overlap well with eBird locations? Birders may be biased towards certain diversity hotspots, and trends in these hotspots may differ from more general locations.

Table1: It is notable how much higher the eBird detection rates are compared to the BBS detection rates. Why is that?

L147: I realize that list length is a wrapper for a lot of different effort variables, but it is worth checking to make sure that additional variation cannot be explained by distance, duration etc. Perhaps concordance would increase with more effort accounted for.

L152-153: Is a linear effect of DOY a reasonable assumption? Could there not be a peak in activity during the breeding season? In which case it might be better to include a quadratic term.

L166-167: It is unclear to me what the difference is between detection probability and the probability of encountering a species on an eBird checklist? Are they not both simply the proportion of checklists on which a given species was detected/encountered?

L172-175: The main difference between the detection probability approach and the encounter rate approach seems to be the models used (GLMM vs random forest) and the variables in those models, rather than the response variables? I am probably missing something but it is unclear.

L192: What distribution was fitted to the detection data?

L201-201: In the first set of comparisons, it does not say what is actually compared: temporal trends, annual estimates, or inter-annual changes? From the results it seems to be the latter two.

L229-230: Where did this p-value come from? I can’t see in the methods what test was used to assess the relationship, only that correlation coefficients were calculated.

Figs. 2-4: The resolution was very low on these figures (a pdf compilation issue I’m sure) making it difficult to really make out what’s what.

L292: Could the results for starlings (here, incorrectly spelled S. vularis) be thrown off by the fact that they are flocking?

L294-295: Why might they differ? It seems crucial to interpret these discrepancies.

L209-304: Could one combine the best aspects of both modeling processes?

L305-306: It is worth noting that eBird data do not do a great job of predicting population trends globally (Neate-Clegg et al. 2020, Biological Conservation).

L348-351: So why not do that?

L362: What should we conclude about which eBird index is best?

Reviewer #2: It is an interesting study that compares four indices of relative abundance of birds obtained from two data sets (eBird and BBS) over several years. The manuscript is technically sound, and the data support the conclusions well. I believe that the statistical analysis was carried out properly and rigorously. Los autores hicieron que todos los datos subyacentes a los hallazgos en su manuscrito estuvieran completamente disponibles.

6. PLOS authors have the option to publish the peer review history of their article (what does this mean?). If published, this will include your full peer review and any attached files.

Reviewer #1: No

Reviewer #2: No

---

## [Author Response · Author response to Decision Letter 0]

10 May 2021

Reviewer 1

0.1 Main Comments

We thank Reviewer 1 for their favorable comments and constructive feedback. We appreciate their suggestions for improving the scope and robustness of our findings. 

Their first comment relates to the selection of our study species. The reviewer correctly states that our study species are limited and have taxonomic bias towards select functional groups known to cause electrical outages. We conducted this analysis of eBird data with the primary focus of assessing its reliability at finer spatiotemporal resolutions for usage in an electrical outage case study. Although our sample size is not large, our study species represent a wide range of factors known to influence detectability and data reliability, and we find relatively consistent results across these species. For example, they cover a broad spectrum of prevalence and detectability (as seen in Table 1), functional distinctness based on life histories such as nesting habitat (cavity, tree, shrub, cliff nesting), foraging strategy (ground, bark, soaring, diving), and diet (insectivore, carnivore, scavenger, granivore), variety in conservation status (invasives such as European Starling, and conservation successes such as Osprey), and occupy a variety of habitats (forest interior, coastal and open water, open habitat, and developed land). 

Nevertheless, we recognize that this is a subset of all possible species and we mean to present this as a case study that highlights the need for increased vigilance as future bird population studies consider various modeling applications of eBird data, rather than an assessment of all available eBird data. We clarified the wording of our study scope in the introduction (lines 46-48; note that line numbers refer to revised version without track changes), methods (lines 79-80), and discussion lines (366-370), and provide further details of the taxonomic bias, but also diversity of detectability in our study species on lines 96-103 and 364-365. 

Following the reviewer's suggestion, we added an analysis of whether local spatial attributes such as habitat cover and eBird survey density would impact the survey accuracy of either dataset and indirectly influence the correspondence of indices between them. We added this as an additional section in our results and discussion looking at finer spatiotemporal resolutions (annual and inter-annual, town-level estimates).

For this, we extracted town-level index predictions for each Massachusetts town (i.e., before averaging them for statewide indices) and calculated Spearman Rank correlations between indices at these finer spatial units. We used BBS detection probability as our BBS relative abundance index, since BBS count and detection yielded similar estimates, and compared it with both eBird indices. We mapped the correlation coefficients between BBS detection probability and each eBird index in each town to visualize the spatial variation in correspondence between these datasets. These maps showed that the degree of correspondence between annual indices and their inter-annual changes varied between towns across the state. The extent of this variance differed between species and the indices being compared. We compiled the maps in an additional supplemental document (S4 Appendix) (lines 496-500). 

We tested the relationship between index reliability and habitat attributes using a linear mixed model. We used the town-level correlation coefficients as a response, and the spatial features of each town including the density of eBird checklist locations, and the proportion of deciduous forest, developed, wetland, herbaceous, and shrub habitat as fixed predictor variables. Our habitat cover types were extracted using the National Land Cover Database's (NLCD) 2016 release and aggregating wetland, forest, developed, and herbaceous land cover classifications. Since we only had spatial locations for the start of each BBS route, this limited our number of spatial samples across the state. To increase the sample size for analysis, we combined the town-level correlation coefficients across species and included species as a random effect. We found that habitat is a driver of correlation between eBird and BBS annual estimates and anticipate it is due to changes in survey accuracy between different habitat types [1-6]. We add these additional questions and methods to the introduction (lines 46-57) and methods (lines 267-286) sections and report our findings in the results (lines 335-354, Figure 4, Table 4) and discussion (lines 443-452).

We address additional clarifications needed in the methods in the minor comments section below.

0.2 Minor Comments

Title: "The title seems a little vague; it doesn't really give a sense of the paper. Perhaps

something like: 'Comparing the reliability of bird abundance indices between

standardized surveys and community science data at fine-scale temporal resolutions'."

We changed the title from "Reliability of relative bird abundance indices at fine-scale

temporal resolutions" to "Comparing the reliability of relative bird abundance indices

from standardized surveys and community science data at finer resolutions."

L31: "The fact that eBird data can capture intra-annual changes in abundance is

mentioned a couple of times, but how can one compare intra-annual abundance changes

between community science data and standardized survey data? The latter have narrow

temporal windows (the breeding season, or Christmas). This is not really an issue the

paper addresses."

The reviewer raises an important issue of finding validation data for finer temporal resolutions (intra-annual changes). Previous studies have addressed this issue by validating estimates with with localized monitoring/survey efforts [10, 11]. These data are typically finer resolution studies at smaller scales and extents. We add note of this validation data issue for future applications of these data on lines 455-459 in the discussion. We broaden our wording when describing the finer resolutions of eBird data in the introduction (lines 24, 30-31) because we do not address these inter-annual trends in our study due to the limitation in available intra-annual validation data.

L58, L64: "Why not 2005-2020? eBird data have increased exponentially, so two years

could make a big difference. Moreover, when estimating temporal trends over such short

time spans, every year counts!"

At the start of this project (April 2020), eBird data were only available through May of 2019, leaving 2018 as the last breeding season with complete data collection for both eBird and BBS. 

L68-69: "How does the reliability of abundance indices vary by habitat?"

Refer to Reviewer 1 Main Comments section, comment 2.

L73: "14 is not many species and, given the aims of this paper, the subset of species

chosen (those that can cause electrical outages) does not seem relevant or representative."

Refer to Reviewer 1 Main Comments section, comment 1.

L101: "This criterion might not be stringent enough. Even if I'm looking at a target

species, I might add a house finch or starling as it flies over. Perhaps >=5 would be a

better cut-off. Horns et al. (reference 18) use >=4."

As suggested, we have adjusted the eBird checklist cut-off to include only checklists with ≥5 species reported (line 115). This reduced the data that were available for our subsequent models and analyses. Species prevalence (frequency of detections) in each dataset increased for all species except house sparrow (these changes were made to Table 1). We adjusted all figures, tables, and supplemental materials with these updated results. Our overall findings did not change, but the correlation coefficients and p-values reported in our results section (lines 292-299) were also updated.

L135-136: "Geographically, did BBS survey locations overlap well with eBird locations?

Birders may be biased towards certain diversity hotspots, and trends in these hotspots

may differ from more general locations."

There were fewer BBS survey locations (route starting locations) (n=25) than eBird survey locations across the state. BBS surveys were also more evenly spatially distributed compared to eBird which tended to be spatially biased near urban areas and along road ways. To see if towns with eBird "hotspots" affected the degree of correspondence between eBird and BBS indices, we also included the relative density of eBird survey locations within each town as a predictor of the correlation between indices (see our response to Main Comment section, comment 2). 

Table1: "It is notable how much higher the eBird detection rates are compared to the

BBS detection rates. Why is that?"

This seems to be a common discrepancy found between these two datasets. Walker and

Taylor (2017) find similar results in their detection rates for multiple study species as

well. We are unsure of the cause for this discrepancy, but sampling frequency and

protocol differ greatly between these datasets which could impact detection rates.

L147: "I realize that list length is a wrapper for a lot of different effort variables, but it is

worth checking to make sure that additional variation cannot be explained by distance,

duration etc. Perhaps concordance would increase with more effort accounted for."

For each species, we compared models that included additional effort variables (e.g., survey distance, duration, number of observers) against models including only list length. Using the R-package AICcmodavg, we compared models using a model selection table that ranked models by the Second-order Akaike Information Criterion (AICc). We found that the latter models out performed accounting for additional effort variables. We added this clarification to the methods section on lines 163-171.

L152-153: "Is a linear effect of DOY a reasonable assumption? Could there not be a peak

in activity during the breeding season? In which case it might be better to include a

quadratic term."

For each species, we compared models that included day of year as a quadratic term, day of year as a linear effect, the untransformed list length and also the log of list length. We compared models using the same model selection table mentioned above. Models that used linear effects of DOY were ranked higher than quadratic effects for all species. List length and log(list length) models varied between species and were selected on a species by species basis. We added this information to the methods section on lines 163-171.

L166-167: "It is unclear to me what the difference is between detection probability and the

probability of encountering a species on an eBird checklist? Are they not both simply the

proportion of checklists on which a given species was detected/encountered?"

The reviewer is correct. These indices both measure the proportion/probability of surveys with species detections. Our intention for these two indices was to test the reliability across two different modeling methods that try to capture the same estimates. We added clarification of this in the methods section on lines 159-160.

L172-175: "The main difference between the detection probability approach and the

encounter rate approach seems to be the models used (GLMM vs random forest) and the

variables in those models, rather than the response variables? I am probably missing

something but it is unclear."

We added clarification in our methods section (lines 159-160) and aims (lines 54-56) that the selection of detection probability and encounter rates as our metrics were to compare two different modeling methodologies that try to capture the same response (the likelihood of detecting a species on a standardized survey). We expected these methods to capture the same response and show similar trends, however our estimates showed moderate to no correlation between these indices' inter-annual trends. We conclude that modeling methodology can impact the reliability of abundance indices, and discuss these points in lines 376-400.

L192: "What distribution was fitted to the detection data?"

We used a binomial distribution for the BBS detection data, similar to the eBird detection data. We added this clarification to the manuscript (line 219).

L201-201: "In the first set of comparisons, it does not say what is actually compared:

temporal trends, annual estimates, or inter-annual changes? From the results it seems to

be the latter two."

Based on the changes in our methods and aims (mentioned above), we restructured our results section to the following order:

Comparing indices within the same dataset at three temporal resolutions (multi-year trends, annual estimates, inter-annual changes) (lines 288-308).

Comparing indices across different datasets at three temporal resolutions (lines 309-325).

Comparing indices across different datasets at finer spatial (town boundaries) and temporal resolutions (annual estimates and inter-annual changes) (lines 335-354).

L229-230: "Where did this p-value come from? I can't see in the methods what test was

used to assess the relationship, only that correlation coefficients were calculated."

To quantify the reliability between relative abundance indices, we calculated correlation coefficients between each pair of indices and assessed their significance using the p-values extracted from the cor.test function in base R. We added this to the methods section on lines 231-232.

Figs. 2-4: "The resolution was very low on these figures (a pdf compilation issue I'm

sure) making it difficult to really make out what's what."

We apologize for difficulties that the figure rendering issues may have caused. We confirmed using PACE (Picture Analysis and Conversion Engine) that our figures comply with the specified requirements by the journal (TIF file format within 2250x2625 pixels at 300 DPI).

L292: "Could the results for starlings (here, incorrectly spelled S. vularis) be thrown off

by the fact that they are flocking?"

In our updated results, the BBS indices for S. vulgaris and A. phoeniceus showed the weakest correspondence between annual estimates and inter-annual changes accordingly (lines 294-299). [7] supports the hypothesis that flocking species (such as S. vulgaris and A. phoeniceus) tend to have higher observational errors and lower accuracy because counting flocks requires greater observer skill. We add this supporting hypothesis to our discussion on lines 373-376. We also correct the spelling for S. vulgaris. 

L294-295: "Why might they differ? It seems crucial to interpret these discrepancies."

We interpret the discrepancies between eBird modeling methods on lines 381-400. We attribute these differences to the data assumptions in the modeling methods themselves (e.g., accounting for hierarchical patterns using GLMMs) and in the amount of data pre-processing used to remove spatiotemporal and sampling bias.

L209-304: "Could one combine the best aspects of both modeling processes?"

We highly recommend using the data preparation best practices outlined in the encounter rate methodology (Strimas-Mackey et al. 2020) to further balance the data before applying it to any model, including the GLMM approach. We add this as a suggestion for future eBird modeling efforts on lines 398-400. 

L305-306: "It is worth noting that eBird data do not do a great job of predicting

population trends globally (Neate-Clegg et al. 2020, Biological Conservation)."

We add this citation [9] to our discussion on line 405 to support our claim that scales and resolutions impact the reliability of predicted trends from eBird data.

L348-351: "So why not do that?"

We intended for our analysis to represent a case study using a subset of focal species. Please see our response to Main Comments section, comment 1 for additional details.

L362: "What should we conclude about which eBird index is best?"

Our results do not show that one eBird index is better than the other. Neither index showed strong correspondence with BBS at finer spatial and temporal resolutions and the degree of correspondence also varied across species. We add mention of this on lines 384-388 and also suggest that each modeling method may offer advantages at different resolutions as supported by their use in previous studies (Walker and Taylor 2017, Horns et al. 2018, Strimas-Mackey et al. 2020) [8].

Reviewer 2

0.3 Main Comments

It is an interesting study that compares four indices of relative abundance of birds

obtained from two data sets (eBird and BBS) over several years. The manuscript is

technically sound, and the data support the conclusions well. I believe that the statistical

analysis was carried out properly and rigorously. Los autores hicieron que todos los datos

subyacentes a los hallazgos en su manuscrito estuvieran completamente disponibles.

We thank Reviewer 2 for their positive comments and support.

Additional references added based on reviewer comments:

1. Rigby EA, Johnson DH. Factors affecting detection probability, effective area

surveyed, and species misidentification in grassland bird point counts. The

Condor. 2019;121(3). doi:10.1093/condor/duz030.

2. Alldredge MW, Simons TR, Pollock KH. Factors affecting aural detections of

songbirds. Ecological Applications. 2007;17(3):948{955.

doi:https://doi.org/10.1890/06-0685.

3. Pacifici K, Simons TR, Pollock KH. Effects of vegetation and background noise

on the detection process in auditory avian point-count surveys. The Auk.

2008;125(3):600{607. doi:10.1525/auk.2008.07078.

4. Diefenbach DR, Brauning DW, Mattice JA. Variability in grassland bird counts

related to observer differences and species detection rates. The Auk.

2003;120(4):1168{1179. doi:10.1093/auk/120.4.1168.

5. Ku laga K, Budka M. Bird species detection by an observer and an autonomous

sound recorder in two different environments: Forest and farmland. PLOS ONE.

2019;14(2):e0211970. doi:10.1371/journal.pone.0211970.

6. Lele SR, Moreno M, Bayne E. Dealing with detection error in site occupancy

surveys: what can we do with a single survey? Journal of Plant Ecology.

2012;5(1):22{31. doi:10.1093/jpe/rtr042.

7. Ganter B, Madsen J. An examination of methods to estimate population size in

wintering geese. Bird Study. 2001;48(1):13.

8. James G, Witten D, Hastie T, Tibshirani R. An introduction to statistical

learning with applications in R. vol. 103 of Springer Texts in Statistics. 1st ed.

New York, NY: Springer Science+Business Media; 2013. Available from:

https://doi.org/10.1007/978-1-4614-7138-7.

9. Neate-Clegg MHC, Horns JJ, Alder FR, Kemahlı Aytekin MC, Şekercioğlu CH.

Monitoring the world's bird populations with community science data. Biological

Conservation;248.

10. Furnas B. Rapid and varied responses of songbirds to climate change in

California coniferous forests. Biological Conservation. 2020;241:108347.

doi:10.1016/j.biocon.2019.108347.

11. Heim W, Heim R, Beermann I, Burkovskiy O, Gerasimov Y, Ktitorov P, et al.

Using geolocator tracking data and ringing archives to validate citizen-science

based seasonal predictions of bird distribution in a data-poor region. Global

Ecology and Conservation. 2020;24:e01215. doi:10.1016/j.gecco.2020.e01215.

---

## [Decision Letter · Decision Letter 1]

22 Jun 2021

PONE-D-20-40357R1

Comparing the reliability of relative bird abundance indices from standardized surveys and community science data at finer resolutions

PLOS ONE

Dear Dr. Feng,

Thank you for submitting your manuscript to PLOS ONE. After careful consideration, we feel that it has merit but does not fully meet PLOS ONE’s publication criteria as it currently stands. Therefore, we invite you to submit a revised version of the manuscript that addresses the points raised during the review process.

We look forward to receiving your revised manuscript.

Kind regards,

Daniel de Paiva Silva, Ph.D.

Academic Editor

PLOS ONE

Journal Requirements:

Additional Editor Comments (if provided):

Dear Feng et al.,

Congratulations. We are almost there. Both reviewers indicated that minor reviews should be performed to your study, and as soon as you complete them, you MS will be accepted for publication in PLoS One.

In this new review round, a new reviewer had to step in. Nonetheless, the suggestions made by him/her were feasible and pertinent, considering this second review round. I believe you will not find any trouble in performing the required changes.

Please do not forget to prepare a rebuttal letter by the time of your resubmission, explaining the changes you did and if needed providing the arguments for improvements that were not able to be done. I will grant you a two-months period (22nd August, 2021) for the completion of the required changes. In case you need more time, please let me know. Do not hesitate to resubmit earlier if you can In case you are able to.

Best regards,

Daniel Silva

Reviewers' comments:

Reviewer's Responses to Questions

**Comments to the Author**

1. If the authors have adequately addressed your comments raised in a previous round of review and you feel that this manuscript is now acceptable for publication, you may indicate that here to bypass the “Comments to the Author” section, enter your conflict of interest statement in the “Confidential to Editor” section, and submit your "Accept" recommendation.

Reviewer #1: All comments have been addressed

Reviewer #3: (No Response)

2. Is the manuscript technically sound, and do the data support the conclusions?

Reviewer #1: Yes

Reviewer #3: Yes

3. Has the statistical analysis been performed appropriately and rigorously? 

Reviewer #1: Yes

Reviewer #3: Yes

4. Have the authors made all data underlying the findings in their manuscript fully available?

Reviewer #1: Yes

Reviewer #3: Yes

5. Is the manuscript presented in an intelligible fashion and written in standard English?

Reviewer #1: Yes

Reviewer #3: Yes

6. Review Comments to the Author

Reviewer #1: I appreciate the thorough job the authors have done revising the manuscript. I’d say that all of my comments have pretty much been addressed. The analyses are very comprehensive and live up to the title, providing a useful piece in the growing eBird literature. I like the addition of the town-level analyses. It’s a shame that one cannot draw stronger conclusions on the utility of eBird data from the study, but it’s important to demonstrate the complexity of relationships between community science and standardized data.

I only have a couple of small comments to make.

L31-33: I know what point is being made here, that eBird checklists are generally reduced to presence/absence. But eBird data do contain counts, and if eBird users submitted more checklists with more counts instead of ‘X’s we’d be able to leverage that for relative abundance. Here I would just clarify that, while eBird checklists can provide count data, they are generally reduced to presence/absence due to gaps in the count data.

L199: I am curious what the reasoning is about including effort in the random forest models but not the GLMMs? Is it that random forests can take all the variables because it assesses their relative importance while in GLMMs you need to choose a best candidate model? Because the two approaches differ in both modelling framework (GLMM vs RF) and model covariates, its difficult to assess whether discrepancies between the estimates result from the former or the latter.

L222: Could list-length analysis be relevant here? BBS participants likely differ in ability, and some may be more likely to detect species than others

L239-240: I am still unsure about distinguishing detection probabilities from encounter rates, although I appreciate the clarification. The authors acknowledge that they “both measure the proportion/probability of surveys with species detections.” I therefore think it is misleading to redefine the two terms when they mean the same thing. Its totally reasonable to use different modelling approaches on the same response variable, and they could be referred to as detection probability GLMM vs detection probability random forest (DP-GLMM vs DP-RF). To think of an analogy from species distribution modelling, often different modelling approaches are compared (e.g. GLMM, GAM, RF, Maxent) on the same response variable (presence/absence). In those instances, you hear them say the GLMM vs the RF, not the presence/absence model vs the occurrence model. At the very least, I think the text should clarify that the two terms are functionally identical and are only differentiated to aid with communication.

Reviewer #3: Review of "Comparing the reliability of relative bird abundance indices from standardized surveys and community science data at finer resolutions".

Feng and Che-Castaldo developed an interesting paper that compares relative abundance metrics from eBird and BBS across different spatial and temporal scales. I agree with the authors in the importance of this topic, as biodiversity monitoring and conservation are increasingly relying on large-scale community science surveys to inform management decisions. Analyses that integrate multiple data sources often do not consider differences in spatial and/or temporal scales, and thus I feel this paper presents an important message to practitioners that differing spatial and temporal scales, as well as modeling approach, cannot be ignored when using large scale data sources.

As I am coming into the review stage at a later point, I would first like to applaud the authors on an exceptional job at responding to the constructive comments of Reviewer 1. The authors made comprehensive edits to their manuscript in response to their comments and clearly communicated this to the reviewer, which together appear to have greatly improved the quality of the manuscript.

I generally find the manuscript clear and informative. My primary concern is with error propagation throughout the analyses, which I feel could have an important impact on the results, especially given the emphasis the authors have placed on statistically significant findings (see general comments below). I also have a few more minor comments.

General Comments

The authors first use four different models to calculate relative abundance indices from eBird and BBS data. The main portion of the work then involves either (1) the use of these indices in correlation tests to assess the consistency in estimates across models and data types; (2) using these indices in simple linear models to estimate temporal trends (i.e., Table 2); or (3) deriving correlation coefficients for these indices and subsequently using the correlation coefficients in linear mixed models to assess effects of habitat type and eBird survey density on these correlations. However, the relative abundance indices have uncertainty associated with them from the model outputs, and it is probably likely the amount of uncertainty varies across the different modeling approaches, species, and data sets. A proper treatment of uncertainty would then propagate the uncertainty in these indices into further analyses that use these results. By treating the relative abundance indices from the model output as known, the authors ignore this uncertainty. This may not have an influence on the overall value of resulting "downstream" statistical tests (e.g., the estimated value of the correlation coefficient or temporal trend likely would not change), but it likely has a large influence on the uncertainty (and thus statistical significance) on these values. This is particularly true for the linear mixed model analysis in lines 342-350, as it does not propagate uncertainty from either the relative abundance indices into the correlation coefficients, or the correlation coefficients into the LMM.

In a frequentist perspective, a bootstrapping approach is a common way to propagate the uncertainty from one analyses to the next. A more simple approach could be to take the relative abundance indices and then weight these values by their standard errors in subsequent linear models that use the relative abundance indices as dependent variables. I believe proper uncertainty propagation from one analysis to the next is necessary to adequately assess the statistical significance of the different patterns the authors are reporting. Performing such an approach would greatly improve my confidence in the underlying results, as well as for many quantitative ecologists that would be interested in this work, and so I'll suggest the authors explore these alternative routes to more accurately account for uncertainty in their analyses. At the very least, the authors should include a paragraph in the discussion describing the lack of uncertainty propagation and the implications this has on the resulting inference.

Specific Comments

- 162-163: Please state the specific algorithm used to fit the GLMMs, which if the default of the glmer function in lme4 was used is Laplace approximation. While this may be clear to R users, this is not as clear to people using other software, and is important given differences among frequentist approximation methods for GLMMs (Bolker et al., 2009).

- Line 194: How was the spatial and temporal subsampling done? Was it done following recommendations in Johnston et al (citation 33)? Clarification of the specific filtering approaches used would be helpful, even if it is just a reference to some other paper.

- Lines 398-400: why did the authors not do this in their analyses? It seems odd to compare one model that uses the best practices for eBird data filtering and another model that does not, because then, as the authors mention, differences between the approaches could be a result of the differences in filtering processes, rather than more of an assessment of the actual modeling approach. I understand they are comparing commonly used methods for modeling eBird data, but the GLMM approach was from a paper in 2017, and the eBird best practices have been updated numerous times since then, so why not update the model with the best filtering practices and then compare it to the random forest model?

References

Bolker, B. M., Brooks, M. E., Clark, C. J., Geange, S. W., Poulsen, J. R., Stevens, M. H. H., & White, J. S. S. (2009). Generalized linear mixed models: a practical guide for ecology and evolution. Trends in ecology & evolution, 24(3), 127-135.

7. PLOS authors have the option to publish the peer review history of their article (what does this mean?). If published, this will include your full peer review and any attached files.

Reviewer #1: **Yes: **Montague Neate-Clegg

Reviewer #3: No

---

## [Author Response · Author response to Decision Letter 1]

21 Aug 2021

Dear Dr. Paiva Silva and our Reviewers,

We thank you for your continued consideration of our manuscript for publication with PLOS ONE as well as the favorable comments and helpful suggestions. Our adjustments based on reviewer comments have improved the the consistency of our methods and overall interpretation of our results. Following this round of feedback, we now apply the same data preparation methods for both eBird indices, instead of strictly following the existing protocols for each method, so that only the modeling methods differ between the indices. This includes filtering the eBird data by effort variables, using the same time window of observations (May-July), sub-sampling to reduce spatiotemporal bias and class imbalance (ratio of detections to non-detections), and balancing sample sizes across years. In addition to data preparation, we also adjusted the covariates used in the two eBird models (Generalized Linear Mixed Models and Random Forests) to include the same effort variables (list length, protocol type, list length * protocol type, time of day, and day of year). We go into more details on these changes in the comments below. 

This standardization allowed us to attribute any differences between the indices to the modeling method rather than inconsistencies in the input data or model covariates. Through this process we also updated our R-script to eliminate the unnecessary step of filling in data for unsurveyed towns, which we realized were generating false non-detections in our model training data. These changes have altered our model performance and results in subsequent analyses, and we have revised our manuscript accordingly. Specifically, the overall RF model performance decreased, so that while AUC values still indicated substantial agreement between model predictions and the testing data, there was only moderate agreement when looking at maximized Kappa (lines 212-214). Although our estimates changed as a result of these model and data adjustments (several species showing improved consistency between annual estimates), our overall findings remained the same: species with strong multi-year trends showed the strongest agreement between annual estimate across datasets and modeling methods. We reported our new findings on lines 312-369 and updated figures and tables along with the abstract and discussion (lines 386-404, 416-428) to reflect the new results. Specifically, we found that Osprey no longer showed consistent estimates across all indices and the eBird GLMM method was on average more consistent with BBS indices. We interpret this last result as evidence that maintaining consistent modeling approaches can improve the reliability of relative abundance estimates across datasets, at least at coarser temporal resolutions.

Reviewer 1

Minor Comments

L31-33: ”I know what point is being made here, that eBird checklists are generally reduced to presence/absence. But eBird data do contain counts, and if eBird users submitted more checklists with more counts instead of ‘X’s we’d be able to leverage that for relative abundance. Here I would just clarify that, while eBird checklists can provide count data, they are generally reduced to presence/absence due to gaps in the count data.”

We added this clarification on the limited availability of eBird count data to lines 32 and 33.

L199: "I am curious what the reasoning is about including effort in the random forest models but not the GLMMs? Is it that random forests can take all the variables because it assesses their relative importance while in GLMMs you need to choose a best candidate model? Because the two approaches differ in both modelling framework (GLMM vs RF) and model covariates, its difficult to assess whether discrepancies between the estimates result from the former or the latter."

We followed the RF methodology described in the eBird Best Practices which includes multiple effort variables, and additionally included list length because it was shown to be representative of multiple effort covariates in previous eBird abundance models. We verified this finding using the RFs ability to assess variable importance. When we assessed the relative importance of each effort variable in the RF, list length was the most important effort variable, consistent across all species. Because of this and based on the reviewer’s comment, we recognize it may be redundant to include the other effort variables as well as list length. We now include only list length and protocol type as effort variables in the RF models (removing observer ID, survey duration, distance, and number of observers). The new model was Detection ∼ Year + Day of Year + Time of Survey + Protocol Type + List Length + Town, and we found little change in model performance compared to models using all effort variables (less than a 3% change in MSE, Sensitivity, Specificity, AUC, Kappa values) for all species models. We now use this same set of predictors in both the eBird RF and GLMM models, which should limit the discrepancies between the RF and GLMM methods to be due only to differences in model type rather than covariate selection. We added all the above mentioned changes to the RF methods section on lines 203-208 and to S1 Appendix which describes the details of our RF method.

L222: "Could list-length analysis be relevant here? BBS participants likely differ in ability, and some may be more likely to detect species than others."

We do not include list length as an effort variable in BBS models because the BBS conducts structured surveys in order to account for survey effort. We do include observer ID as a random effect to take into account any additional differences between observer skill. We make note of this on lines 224-227.

L239-240: "I am still unsure about distinguishing detection probabilities from encounter rates, although I appreciate the clarification. The authors acknowledge that they 'both measure the proportion/probability of surveys with species detections.' I therefore think it is misleading to redefine the two terms when they mean the same thing. Its totally reasonable to use different modelling approaches on the same response variable, and they could be referred to as detection probability GLMM vs detection probability random forest (DP-GLMM vs DP-RF). To think of an analogy from species distribution modelling, often different modelling approaches are compared (e.g. GLMM, GAM, RF, Maxent) on the same response variable (presence/absence). In those instances, you hear them say the GLMM vs the RF, not the presence/absence model vs the occurrence model. At the very least, I think the text should clarify that the two terms are functionally identical and are only differentiated to aid with communication."

We thank the reviewer for their help clarifying our terminology for readers. We make their suggested change and instead of referring to RF estimates as ”encounter rates” and GLMM estimates as ”detection probabilities” we refer to both by their model frameworks rather than their outputs. All instances of ”encounter rate” and ”detection probability” in the manuscript text, tables, and figures were adjusted to ”detection probability” using either the RF or GLMM method (abbreviated as ”DP-RF” and ”DP-GLMM”).

Reviewer 3

Main Comments

”The authors first use four different models to calculate relative abundance indices from eBird and BBS data. The main portion of the work then involves either (1) the use of these indices in correlation tests to assess the consistency in estimates across models and data types; (2) using these indices in simple linear models to estimate temporal trends (i.e., Table 2); or (3) deriving correlation coefficients for these indices and subsequently using the correlation coefficients in linear mixed models to assess effects of habitat type and eBird survey density on these correlations. However, the relative abundance indices have uncertainty associated with them from the model outputs, and it is probably likely the amount of uncertainty varies across the different modeling approaches, species, and data sets. A proper treatment of uncertainty would then propagate the uncertainty in these indices into further analyses that use these results. By treating the relative abundance indices from the model output as known, the authors ignore this uncertainty. This may not have an influence on the overall value of resulting ”downstream” statistical tests (e.g., the estimated value of the correlation coefficient or temporal trend likely would not change), but it likely has a large influence on the uncertainty (and thus statistical significance) on these values. This is particularly true for the linear mixed model analysis in lines 342-350, as it does not propagate uncertainty from either the relative abundance indices into the correlation coefficients, or the correlation coefficients into the LMM. In a frequentist perspective, a bootstrapping approach is a common way to propagate the uncertainty from one analyses to the next. A more simple approach could be to take the relative abundance indices and then weight these values by their standard errors in subsequent linear models that use the relative abundance indices as dependent variables. I believe proper uncertainty propagation from one analysis to the next is necessary to adequately assess the statistical significance of the different patterns the authors are reporting. Performing such an approach would greatly improve my confidence in the underlying results, as well as for many quantitative ecologists that would be interested in this work, and so I’ll suggest the authors explore these alternative routes to more accurately account for uncertainty in their analyses. At the very least, the authors

should include a paragraph in the discussion describing the lack of uncertainty propagation and the implications this has on the resulting inference.”

We thank Reviewer 3 for their feedback and agree that it would be important to account for uncertainty propagation in our analyses. To address this, we first calculated standard errors (SEs) for each town-level estimate. For the GLMMs, we used the bootMer function from R-package lme4 to calculate standard errors for the town-level estimates using bootstrapping. For the RF models, we directly extracted SEs from the prediction function (se.fit = TRUE in predict.lm). We used the reciprocals of these SEs as weights to calculate weighted correlation coefficients for town-level estimates. We used the cor.wt function from the R-package psych to implement weighted Pearson correlations. This placed greater weight on estimates with smaller standard errors. We then use these weighted correlation coefficients in our analysis examining whether the correspondence between eBird and BBS indices depended on habitat type and eBird survey density. To incorporate uncertainty into our state-level analyses, we weighted the town-level

estimates with the reciprocals of their SEs, and then averaged the weighted estimates across all towns in the state. We added these methods to lines 233-243.

We address additional clarifications needed in the methods in the minor comments section below.

Minor Comments

L162-163: ”Please state the specific algorithm used to fit the GLMMs, which if the default of the glmer function in lme4 was used is Laplace approximation. While this may be clear to R users, this is not as clear to people using other software, and is important given differences among frequentist approximation methods for GLMMs (Bolker et al., 2009).”

We used a modified Laplace approximation method in which the random effects and fixedeffects coefficients were optimized in the penalized iteratively reweighted least squares step rather than in the nonlinear optimizer (parameter nAGQ set to 0 in function glmm), which allowed our models to converge. We added Bates et al. (2021) and Bolker et al. (2009) to our references and clarify our GLMM approximation method on lines 174-177.

L194: ”How was the spatial and temporal subsampling done? Was it done following recommendations in Johnston et al (citation 33)? Clarification of the specific filtering approaches used would be helpful, even if it is just a reference to some other paper.”

The spatial and temporal sub-sampling followed recommendations from Johnston et al. and Strimas-Mackey et al. (2020). We added these references to the appropriate lines (125-127) and refer to S1 Appendix where these specific sub-sampling and filtering approaches are described in more detail.

L398-400: ”Why did the authors not do this in their analyses? It seems odd to compare one model that uses the best practices for eBird data filtering and another model that does not, because then, as the authors mention, differences between the approaches could be a result of the differences in filtering processes, rather than more of an assessment of the actual modeling approach. I understand they are comparing commonly used methods for modeling eBird data, but the GLMM approach was from a paper in 2017, and the eBird best practices have been updated numerous times since then, so why not update the model with the best filtering practices and then compare it to the random forest model?"

We recognize that using the latest best practices for data management is essential for making reliable models, so we now apply the data filtering from the eBird best practices to process the dataset going into both the RF and GLMM models. This includes filtering by the effort variables, as well as sampling the data to reduce spatial and temporal bias and class imbalance in the presence and absence observations and to balance the number of surveys across years. We added this to the methods section on lines 125-127 and to the S1 Appendix and removed this as a suggestion in the discussion.

---

## [Decision Letter · Decision Letter 2]

27 Aug 2021

Comparing the reliability of relative bird abundance indices from standardized surveys and community science data at finer resolutions

PONE-D-20-40357R2

Dear Dr. Feng,

We’re pleased to inform you that your manuscript has been judged scientifically suitable for publication and will be formally accepted for publication once it meets all outstanding technical requirements.

Kind regards,

Daniel de Paiva Silva, Ph.D.

Academic Editor

PLOS ONE

Additional Editor Comments (optional):

Dear Feng et al.,

It is with great pleasure that I announce that yur manuscript is formally accepted for publication in PLoS One! Well done!

Sincerely,

Daniel Silva, Ph.D.

Reviewers' comments:

Reviewer's Responses to Questions

**Comments to the Author**

1. If the authors have adequately addressed your comments raised in a previous round of review and you feel that this manuscript is now acceptable for publication, you may indicate that here to bypass the “Comments to the Author” section, enter your conflict of interest statement in the “Confidential to Editor” section, and submit your "Accept" recommendation.

Reviewer #3: All comments have been addressed

2. Is the manuscript technically sound, and do the data support the conclusions?

Reviewer #3: Yes

3. Has the statistical analysis been performed appropriately and rigorously? 

Reviewer #3: Yes

4. Have the authors made all data underlying the findings in their manuscript fully available?

Reviewer #3: Yes

5. Is the manuscript presented in an intelligible fashion and written in standard English?

Reviewer #3: Yes

6. Review Comments to the Author

Reviewer #3: The authors have done a thorough job of addressing all of my comments and concerns. Congratulations on a very useful contribution to the literature.

7. PLOS authors have the option to publish the peer review history of their article (what does this mean?). If published, this will include your full peer review and any attached files.

Reviewer #3: No

---

## [Editor Report · Acceptance letter]

2 Sep 2021

PONE-D-20-40357R2 

Comparing the reliability of relative bird abundance indices from standardized surveys and community science data at finer resolutions 

Dear Dr. Feng:

I'm pleased to inform you that your manuscript has been deemed suitable for publication in PLOS ONE. Congratulations! Your manuscript is now with our production department. 

Kind regards, 

on behalf of

Dr. Daniel de Paiva Silva 

Academic Editor

PLOS ONE